# Compression and Stretching of Confined Linear and Ring Polymers by Applying Force

**DOI:** 10.3390/polym13234193

**Published:** 2021-11-30

**Authors:** Wenduo Chen, Xiangxin Kong, Qianqian Wei, Huaiyu Chen, Jiayin Liu, Dazhi Jiang

**Affiliations:** 1School of Materials, Sun Yat-sen University, No. 135, Xingang Xi Road, Guangzhou 510275, China; weixx8@mail2.sysu.edu.cn (Q.W.); chenhy97@mail2.sysu.edu.cn (H.C.); liujy265@mail.sysu.edu.cn (J.L.); jiangdzhi@mail.sysu.edu.cn (D.J.); 2School of Materials, Shenzhen Campus of Sun Yat-sen University, No. 66, Gongchang Road, Guangming District, Shenzhen 518107, China; 3Changchun Institute of Applied Chemistry, Chinese Academy of Sciences, Changchun 130022, China; bnnkong@foxmail.com

**Keywords:** ring polymer, compression, confinement, applying force, helical structure

## Abstract

We use Langevin dynamics to study the deformations of linear and ring polymers in different confinements by applying compression and stretching forces on their two sides. Our results show that the compression deformations are the results of an interplay among of polymer rigidity, degree of confinement, and force applied. When the applied force is beyond the threshold required for the buckling transition, the semiflexible chain under the strong confinement firstly buckles; then comes helical deformation. However, under the same force loading, the semiflexible chain under the weaker confinement exhibits buckling instability and shrinks from the folded ends/sides until it becomes three-folded structures. This happens because the strong confinement not only strongly reduces the buckling wavelength, but also increases the critical buckling force threshold. For the weakly confined polymers, in compression process, the flexible linear polymer collapses into condensed states under a small external force, whereas the ring polymer only shows slight shrinkage, due to the excluded volume interactions of two strands in the crowded states. These results are essential for understanding the deformations of the ring biomacromolecules and polymer chains in mechanical compression or driven transport.

## 1. Introduction

Polymer chains are frequently stretched by imposed external constraints, such as confinement or an external force, which are geometric and tensile constraints, respectively, [1]. The synergistic effects of these constraints are encountered in a variety of fields, both academic and technical. Examples include the stretching of polymer melts or concentrated solutions [2,3], force-spectroscopy [4] using optical or magnetic tweezers [5,6], genome mapping [7], and nanopore sequencing [8]. Similarly, the above nanoconfinement techniques are also associated with the compression of polymers by forces or flow; examples are the entry of polymer chains into a microchannel [1], the translation of a polymer through nanopores or entropic trap arrays [9], and DNA injection of a bacteriophage into a host [10,11]. Although the effects of either confinement or an external force on the extension of the polymer chain are understood [1], the simultaneous exist of confinement and tension has gotten less attention.

When a single linear polymer is trapped in a nanochannel, it is extended by the effect of confinement. The degree of confinement is determined by the competition between channel size and chain rigidity [12,13]. According to the degree of confinement, the confined regions are divided into three regions: the de Gennes regime for a weakly confined flexible polymer, the Odijk regime for a strongly confined semiflexible chain, and the transition regime between weak and strong confinement limits. In the de Gennes regime, a polymer chain can be considered as a string of blobs segregated by excluded volume interactions, where the channel size *D* is much larger than the persistent length *Lp* (*D >> Lp*). In the Odijk regime (*D << Lp*), the chain segments are frequently deflected by the channel walls, and all of them are strongly aligned. Between the de Gennes regime and the Odijk regime is the transition regime, where the segments near the walls are aligned but those far from the walls are randomly orientated [14].

As in channel confinement, applying an external force to a polymer chain also separates the linear chain into three regimes, according to the characteristic length $\xi =k_{B}T/f$, where $k_{B}T$ is the thermal energy and *f* is the applied force [15,16]. When the external force overcomes the thermal fluctuation, a polymer chain is separated into a string of tensile blobs. Inside the blobs, the influence of the external force is overwhelmed by thermal fluctuation. This is the classical Pincus regime with ξ>b. *b* is the thermal blob size where free energy from excluded volume interaction equals $k_{B}T$. As the external force increases, the polymer chains deform into the extended Pincus regimes Lp<ξ<b with ideal-chain behavior inside blobs, which has been observed in the stretching experiments of PEG [17]. In the third regime, polymer chains are highly stretched with ξ<Lp, and all segments are unidirectionally oriented, because the segments cannot fold back for when highly stretched. The polymer chains in a semidilute solution are highly similar to those in confinement, in that both cases can be described by the blob model.

Differently from the process of extending polymer chains, the compression processes have received much less attention. Several studies have considered the compression of confined DNA in microchannels via mechanical force [18,19,20,21], electric fields [22,23], and flow fields [7,10], and the results show that the compression deformations are strongly dependent on the chain rigidity, the degree of confinement, and the external force. J. Pelletier et al. first used an optically trapped bead as a micropiston inside a microchannel to compress an Escherichia coli chromosome [7] via weak confinement (Lp/D≈ 50 nm/1.5 μm = 1/30). Benefiting from the recent advances in nanoscale fabrications and single-molecule macromolecular experiments, they found that chromosomes are condensed randomly to develop dense piles of blobs. A. Khorshid and their coworkers pointed out that a weakly confined DNA molecule (Lp/D≈ 50 nm/300 nm = 1/6) is partially compressed at a fast sliding speed [18], whereas the chain exhibits the mean-field behaviors under a slow sliding speed, reflected by the compression deformations of local blobs. Jun et al. established the force–displacement $f-L_{x}$ functions of $fD=-A\left(L_{x}/L_{0}\right)+B\left(L_{x}/L_{0}\right)-2$ for the weak compression regime and $fD∼-Lx^{-9/4}$ for the strong-compression regime, where A and B are constants, *f* is the applied force, $L_{x}$ is the displacement of the chain along the channel, and $L_{0}$ is the equilibrium size [10]. Y. Hayase et al. studied the compression process of a semiflexible chain in strong confinement $L_{p}/D=60$ by adding a slipping wall [24]. They found that the semiflexible chain undergoes a sequence of recurring structural transitions: random deflection along the channel, a helix going around the channel wall, double-folded random deflection, double-folded helix, etc. [25]. Cifra et al. studied polymer chains from strong compression to full stretching with Monte Carlo simulations and established f−R functions [25]. A rigid chain experience buckling instability, such as Euler buckling, when the external force is above a threshold [26,27]. Gompper and their coworkers used mesoscale hydrodynamic simulations to study the semiflexible polymer under the flow-induced compression, and found that the semiflexible polymer undergoes a rod-to-helix transition due to a nonequilibrium and nonstationary buckling transition [28]. Recently, Chakrabarti and their coworkers proved that flexible filaments buckle into helical shapes in strong compression flows in experiments using fluorescently labeled actin filaments in microfluidic divergent flows [29].

The above studies focused on linear polymers. The deformations and dynamics of confined ring polymers in the compression and stretching processes have remained elusive, though the ring polymer is one of the common forms in synthetic chemistry and living beings [30,31,32,33]—e.g., cyclic ethers, plasmid, genome, actin, and polyose. Compared with linear chains, ring polymers have more complex conformation and relaxation states, due to having no ends and strong excluded volume interactions between the two strands [34,35,36].

We study linear and ring polymers with different rigidities trapped in a microchannel and undergoing compression and stretching processes under a constant external force. We focus on confined polymers in two regimes: the weakly confined flexible polymers in the de Gennes regime and the strongly confined semiflexible or rigid chains in the Odijk regime. The flexible coarse-grained model represents PE, PP, PEO, POM, etc.; and the semiflexible chain represents PET, PEEK, DNA, RNA, etc. [37] In Section 2, we introduce Langevin dynamics and coarse-grained models in which linear and ring polymers are compressed and stretched by a couple of external forces in a tube channel. In Section 3, we describe the deformations and dynamics of the confined linear and ring chains during the compression and stretching processes, and investigate the influences of chain rigidity and force magnitude. Finally, we draw conclusions on the mechanisms and the processes of deformations in the Conclusions section.

## 2. Model and Simulation Methods

In this work, a single linear or ring polymer chain is dynamically stretched or compressed in a tube-like nanochannel by an external force, as shown in Figure 1. The external force *f* is applied to two ends of linear polymers (Figure 1a), or on two sides of ring polymers with the maximum distance along the tube channel (Figure 1b).

The conformational and dynamical properties of polymer chains are updated by the Langevin dynamics simulation. The coarse-grained polymer chains with different rigidities are described by the general bead-spring models. The total energy $U_{total}$ includes the excluded-volume interactions $U_{LJ}$, the bonding energy $U_{b}$, the bending potential $U_{\theta }$, the repulsive interactions from the wall of the nanochannel $U_{wall}$, and the extension energy from external force $U_{f}$. The simulation details are as follows: The linear or ring polymer is modeled as a generic semiflexible bead-spring chain with *N* beads of mass *M* (M=1.0). The excluded-volume interactions between particles are considered by a truncated and shifted Lennard–Jones potential $U_{LJ}$ [38,39]:(1)$$U_{LJ}=\left\{\begin{array}{cccc} 4\varepsilon [{\left(\sigma /r\right)}^{12}-{\left(\sigma /r\right)}^{6}]+\varepsilon & \; & & r\leq r_{cutLJ}\\ 0 & \; & & r>r_{cutLJ} \end{array}\right.$$
where *r* denotes the spatial distance between particles *i* and *j*. The parameters ε and σ are taken as the units of energy and length (ε=1 and σ=1), respectively. The short-range, purely repulsive interactions are considered by choosing $r_{cutLJ}=2^{1/6}\sigma$.

The bond connection between the adjacent beads is maintained by Hooke’s law:(2)$$U_{b}\left(r\right)=1/2K_{b}{\left(r-r_{0}\right)}^{2}$$
where the equilibrium bond length $r_{0}$ equals the unit length σ, and the spring constant $k_{b}$ is 10,000 $\varepsilon /\sigma^{2}$. The high spring constant is to ensure small fluctuation of bond length, which is less than 0.1 [40].

The variation of rigidity is imposed by a bending potential $U_{\theta }$ defined as
(3)$$U_{\theta }=K_{\theta }{\left(\cos\theta -\cos\theta_{0}\right)}^{2}$$
where θ is the bond angle, and $\theta_{0}$ is the equilibrium bond angle, set as $180^{\circ }$. $K_{\theta }$ is the strength of harmonic potential. To eliminate chain length effect, the contour lengths *L* of the linear chain is chosen as 100σ and the ring chain 200σ. To study the influence of the chain rigidity, the value of $K_{\theta }$ is set as 0∼1000. The intrinsic persistent length $L_{p}$ is routinely fitted to the relationship $K_{\theta }/k_{B}T=L_{P}/r_{0}$, with $k_{B}T=\epsilon =1.0$.

The flexible and semiflexible ring polymers are confined in a tube channel with diameter d=5σ and d=10σ. The interactions between the rings and the surface of the tube channel $U_{wall}$ are implemented by the truncated and shifted Lennard–Jones potential $U_{LJ}$ with $r_{cutLJ}=2^{1/6}\sigma$.

A pair of external force *f* is applied to the two ends of a single linear polymer or two sides of an individual ring polymer in the opposite directions. The extended chain undergoes compression and stretching processes due to the strong external force. The energy $U_{f}$ represents the product of the force magnitude *f* and the differential of the displacement from the equilibrium chain size, $U_{f}=-f\left(L_{x}-L_{0}\right)$. The external force *f* is in the range from 0.0 ε/σ to 20.0 ε/σ. Considering the free energy cost $∼k_{B}T$ per blob, the external force *f* is normalized by $k_{B}T$. The characteristic length $\xi =k_{B}T/f=1.0$, and the subchains inside the blob followed the Flory behaviors. The trajectory of a ring polymer evolved with time according to Langevin equation:(4)$$md^{2}r_{i}/dt^{2}=-\nabla_{i}U_{total}-\Gamma dr_{i}/dt+W_{i}\left(t\right).$$

$r_{i}$ is the position of the *i*th bead and Γ is the friction coefficient. The random force $W_{i}\left(t\right)$ is Gaussian white noise with zero mean, and the covariance
(5)$$\langle W_{i\alpha }\left(t\right)W_{j\beta }\left(t^{\prime }\right)\rangle \geq \delta_{\alpha \beta }\delta_{ij}\delta \left(t-t^{\prime }\right)2k_{B}T\Gamma$$
where α, β represent *x*, *y*, or *z*. The friction coefficient Γ=m/τ=1.0, with time unit $\tau =\sigma \sqrt{\left(m/\tau \right)}=1.0$. We numerically integrate the Langevin equation with time step δt=0.005τ.

For the initialization, the systems are at equilibrium of 6,000,000 steps to relax the polymer chains. Then a pair of external forces are added to two sides of a chain to stretch and compress it. In the stretching process, each simulation moves 2,000,000 steps to reach a steady state. In the collapsing process, each simulation runs 10,000,000 steps. Under a pair of compression forces applied on the two ends/sides, the polymer chain is firstly compressed from the ends until it reaches the minimum size along the direction of external force. After that, the polymer chain is stretched from the condensed state, as shown in the Appendix A. We chose the states with the minimum sizes in the directions of the respective external forces as the finally collapsed states. We runs 5 parallel samples to carry out statistical analysis.

## 3. Results and Discussion

### 3.1. Semiflexible Polymers in Strong Confinement

The semiflexible polymers in the strong confinement (the Odijk regime) under different compression forces shows complex deformations. Figure 2 shows that the linear and ring polymers with rigidity Lp=100 are trapped in the nanochannels with the diameter d=5. When the external force applied on two sides $f/k_{B}T$ is stronger than the thermal fluctuations, the semiflexible chains initially responds via compressing longitudinally along the axis direction, as in Euler buckling. For the strong confinement, the high-amplitude lateral deflections are suppressed by the channel walls, and the backbone is reflected from the wall; thus, the semiflexible chain undergoing Euler buckling has a shorter wavelength. The force and the wavelength follow the relationship $f∼k_{b}{\left(n\pi /l\right)}^{2}$, with the length along the axis direction *l* and the wave vectors q=nπ/l, where n=1,2,3,…. Under the applied force $f/k_{B}T=1.0$, the semiflexible linear polymer in the strong confinement (d=5) starts to buckle from the two ends and gradually forms helical structures (see Figure 2a). It is interesting that the semiflexible ring prefers to form a double helix in Figure 2c, due to the strong excluded volume interactions of the two strands.

Different from the unconfined states, the strong confined chain was reflected from the wall; thus, the semiflexible chain undergoing Euler buckling has a shorter wavelength λ, which is much smaller than the persistent length. The response of the chain is largely dictated by the bending energy, and the influence of thermal fluctuations can be ignored, as fluctuations are out of the wavelength of buckling. However, thermal fluctuations play an important role in the buckling transitions of unconfined polymers. The buckling transition exhibits a marked change from distinct second-order in the absence of fluctuations to being a smooth transition in the presence of fluctuations [41].

When the force is increased beyond the threshold of the buckling transition $k_{b}{\left(n\pi /l\right)}^{2}$, the chain could not be further compressed for the strong bending energy and the high curvature of the chain. The semiflexible chain undergoes buckling instability, and any additional load would have caused the chain to completely collapse. As shown in Figure 2b,d, under the applied force $f/k_{B}T=10.0$, the chain folds back from the two ends with J-loop or hairpin shape, and finally shrinks to the shortest length with three-folded conformations. The deformation of the semiflexible ring polymer with Lp=100 under a couple of two constant forces is similar to that of the semiflexible linear polymer, though the ring has transition states that includes a double helix for the excluded volume interactions between the two strands.

The inherent structures and spatial correlations are analyzed by the bond correlation function $C_{b}$:(6)$$C_{b}=\langle u_{i}\left(t\right)\cdot u_{j}\left(t\right)\rangle$$
where $u_{i}$ is the unit vector’s tangent to the chain contour at the position of the *i*th segment $u_{i}=\left(r_{i+1}-r_{i}\right)/\left|r_{i+1}-r_{i}\right|$. The bond correlation functions $C_{b}$ of semiflexible linear and ring polymers in their confined states are shown in Figure 3. In the initially uncompressed state, the bond correlation function $C_{b}$ of the linear polymer shows exponential decay, whereas $C_{b}$ of the ring polymer shows a symmetric shape at N/2 because of the closed topology. $C_{b}$ decays to 0 for the semiflexible linear polymer, and declines to the negative minimum at N/2 for the semiflexible ring polymer [42]. It is evident that the semiflexible ring has a positive short-range correlation around N〈l〉/L<0.25, and a negative long-range correlation at N〈l〉/L>0.25, due to the special topology of no ends.

The $C_{b}$ for the strongly confined linear polymers under the compression force $f/k_{B}T=1.0$ is shown in Figure 3a. The semiflexible linear polymer has linear decay to 0 for the initial equilibrium states. When compressed by the applied force, the linear polymer softens under the compression, and it is buckling from a randomly deflected state to a helix structure, with an obviously periodic correlation of $C_{b}$. As the compression increases, the linear polymer forms a perfect and stable helix, and $C_{b}$ decays to 0 with damped oscillations.

The strongly confined semiflexible ring displays linear decay toward to the minimum value in the center, and the two declining regimes are “V” shaped at the initial equilibrium, in Figure 3c. That is all due to the U-turn at the end of the stretched ring being confined in the channel. As compression force f=1.0 is applied, the curve of $C_{b}$ shows dampened oscillation for the formation of double helical structures. Buckling is the first step in the formation of more complex structures, and it is followed by a helical deformation of the polymer. Compared with the linear polymer, the curve of $C_{b}$ for the ring polymer has large thermal fluctuations, because of the strong excluded volume between the two polymer fragments along the ring.

Under the compression force $C_{b}$, the two ends of semiflexible linear polymers are folded back, and polymer segments give evidently negative correlations with the folded structures, as shown in Figure 3b. For when the linear polymer shrink to the three-folded structures, the $C_{b}$ has a valley around 〈N〉/3 and a peak at 2〈N〉/3. Similarly to the linear polymers, the strong confined ring shows two valleys and two peaks for symmetry due to the three-folded structures, as shown in Figure 3d.

When the semiflexible linear and ring polymers are confined in the channel with d=10, the polymer chains are folded back under the compression force $f/k_{B}T=1.0$ without the helix structures, as shown in Figure 4. In the stronger confinement d=5, the high-amplitude lateral deflections are suppressed by the confinement. However, the wave-length *l* is much larger under the weaker confinement d=10; correspondingly, the threshold compression force of buckling instability $f∼k_{b}{\left(n\pi /l\right)}^{2}$ becomes smaller. Thus, the semi-flexible linear and ring polymers confined in channels with d=10 under the compression force $f/k_{B}T=1.0$ undergo folding back from two sides into a “U” or “J” shape, and finally collapse to the three-folded structures.

### 3.2. Flexible Polymers in Weak Confinement

In response to the external force, the weakly confined flexible polymers undergo different deformations from the semiflexible polymers under strong confinement. The flexible polymers confined in the microchannel can be considered strings of isometric blobs (de Gennes regime) or anisometric blobs (extended de Gennes regime). These blobs are segregated by excluded volume interactions, and their size equals the channel diameter *D*.

When the constant stretching force *f* is applied, the polymer chains are extended under the synergistic effects of confinement and stretching [25,43]: the channel induced longitudinal prestretching and the mechanical force pulls the chain. Under the weak force, the blobs are separated into the tensile blobs. When the strong force is applied, the polymers are fully stretched to nearly the contour length. The deformations and dynamics of the weakly confined ring are in agreement with those of the linear polymers qualitatively.

When a couple of compression forces *−f* are applied to the two ends, the weakly confined linear polymer (d=5) starts to shrink from the folded ends until it becomes the three-folded structure in the direction of the external forces, as shown in Figure 5a,c. In the collapsing process, the two ends of the linear polymers are stretched to nearly straight under ($f/k_{B}T=1.0$) with only small transverse fluctuations, but the structures of other beads in the center undergo no obvious changes because of the short time to relax, as shown in Figure 5a. Differently from the straightly stretched ends, the two ends of linear polymer under $(f/k_{B}T=0.01$) maintain the blob states shown in Figure 5b. This is because the influence of the external forces is overwhelmed by thermal fluctuations, and the segments inside each tensile blob behave as they did in the absence of force.

Similarly to the flexible linear polymer, each flexible ring is also compressed from its two sides under a constantly external force ($f/k_{B}T=1.0$), as shown in Figure 5c. It should be noted that the whole ring is aligned along the tube axis due to the condensed states within the three-folded structure. It is surprising that, differently from the linear polymers, the ring polymers undergo less obvious compression under the same weak force ($f/k_{B}T=0.01$), as shown in Figure 5d. The two sides of ring polymers cannot be further compressed, due to the strong excluded volume between the two strands of each ring polymer. It means that, in the compression process, the applied force must be large enough to overcome the excluded volume interactions.

The bond correlation functions $C_{b}$ of the weakly confined flexible chains in the compression processes are shown in Figure 6. The $C_{b}$ displays fast attenuation and decays to 0 in the initial equilibrium states. When compressed by applied force ($f/k_{B}T=1.0$), the flexible linear polymer in weak confinement (d=5) is folded back from two ends, as reflected in the positive correlation at |i−j|>80 and negative correlation at 20<|i−j|<80, as shown in Figure 6a. With the compression increasing, the linear polymer collapses into the three-folded structure, with an obvious maximum value of positive correlation at 2N/3 and a minimum value of negative correlation around N/3. When compressed by applied force ($f/k_{B}T=0.01$), the flexible ring also folds back under the external load. However, the curves of $C_{b}$ show large fluctuations due to the effect of thermal fluctuation (Figure 6b).

The bond correlation function $C_{b}$ of the strongly confined semiflexible ring displays linear decay with a “V” shape in the initial equilibrium, as shown in Figure 6c. As the compression force ($f/k_{B}T=1.0$) is applied, $C_{b}$ follows a symmetric curve, which has a symmetrical axis at N/2. There is a maximum value of positive correlation and a minimum value of negative correlation in the folding-back process. However, for compression force ($f/k_{B}T=0.01$), the curves of $C_{b}$ show no obvious change. This means that the ring polymer cannot be compressed, due to the strong excluded volume between the two strands.

When the flexible linear and ring polymers are confined in the channel with d=10, the polymer chains are folded back under the compression force ($f/k_{B}T=1.0$) without forming helical structures, as shown in Figure 7. The linear polymer is folded back from the two ends. The two ends are stretched out straightly while the rest remains the same during the short relaxation, as shown in Figure 7a. Similar results are found for the flexible ring polymer under ($f/k_{B}T=1.0$), as shown in Figure 7b. The two sides are folded back under external loads into straightened structures, while the rest maintains its blob structures. Finally, the flexible ring collapses into three-folded structures.

### 3.3. Trajectories of Polymer Deformation

In order to deeply understand the stretching and compression processes of the polymer chains with different degrees of confinement, we present the trajectories of deformation for an individual chain with temporal resolution. Figure 8 shows the trajectories of chain length $L_{x}$ as functions of time *t* when the chains with different rigidities are confined in channel diameter d=5. The length $L_{x}$ is the longest dimension of a polymer chain in the direction of channel axis, defined as $L_{x}=max\left(x_{i}\right)-min\left(x_{j}\right)$, where $x_{i}$ and $x_{j}$ are the *x*-coordinates for particles *i* and *j*.

For the weakly confined polymers in the de Gennes regime, the flexible chain is extended under the stretching force and $L_{x}$ reach a plateau, as shown in Figure 8a,d. As the external force increased, the chains are stretched to nearly the full contour length (L=100σ). In the compression process, the linear polymers shrink into condensed states with $L_{x}$ around 22. For $f/k_{B}T=0.01$, the compression length $L_{x}$ is slightly larger than 22, because the strength of the excluded volume interactions within the compressed states is stronger than the applied force. The ring polymer shrinks to $L_{x}\approx 60$, much larger than linear polymers, due to the excluded volume interactions between two strands in the crowded environment, as shown in Figure 8d.

For the strong confined polymers ($k_{b}=100$) in the Odijk regime, the semiflexible chains show only slight stretching of their extended structures in the confined channel, as shown in Figure 8c,f. As the external force increased, the chains are stretched to nearly the full contour length (L=100σ). In the compression process, the linear polymer undergoes buckling and shrinks into helix structures with a constant value. For $f/k_{B}T=10.0$, the compression force overwhelms the bending energy and the excluded volume interactions; thus, the chain collapses into the three-folded structures. Similarly to the linear polymers, the ring polymer is also compressed into double helix structures, as shown in Figure 8f. The length $L_{x}$ of the ring polymer is less than that of the linear polymer due to the excluded volume interactions between the two strands in the nanochannel. Under the stronger force $f/k_{B}T=10.0$, the ring polymers are also compressed into the three-folded structures. Between the Odijk regime and de Gennes regime is the transition regime $(k_{b}=5$), as shown in Figure 8b,e. The semiflexible polymers in this regime show crossover behaviors such that they form helical structures under small force $f/k_{B}T=0.1$, and then they are compressed when the force overwhelms the critical threshold $f/k_{B}T>10.0$. Finally, the semiflexible polymers are also compressed into the three-folded structures.

Figure 9 shows that the polymer chains with different rigidities are stretched and compressed in the confined channel d=10. The flexible polymers are confined in the microchannel composed of a string of isometric blobs, which are separated by excluded volume interactions. Under the external force, they are stretched or compressed, as shown in Figure 9a,d. The semiflexible linear polymer $(k_{b}=100$) in the channel with d=10 has no obvious buckling. This is because compared with the weak confinement, the strong confinement not only strongly reduces the buckling wavelength, but also increases the critical buckling force threshold. The ring shows buckling at $f/k_{B}T=0.1$, as shown in Figure 9f, because of the stronger excluded volume interactions between the two strands. In the transition regime, the polymer chains are compressed into three-folded structures, except for a little shrinkage at $f/k_{B}T=0.01$, where the weakly confined chains maintains the blob conformations.

## 4. Conclusions

We simulate the compression and stretching of linear and ring polymers with different rigidities by adding a couple of external forces on two sides of each chain. The simulation results show that the deformations and dynamics are dependent on the combined interactions of confinement, bending energy, and applied force.

When the force applied is beyond the threshold of buckling transition, the semiflexible chain in the strong confinement (d=5) buckles in the first step and then experiences helical deformation. The semiflexible linear polymer in strong confinements are compressed into a helical structure, and the ring chain forms the double helix. However, under the same force loading, the semiflexible chains under confinement with a channel diameter d=10 exhibit buckling instability and shrink from the folded ends/sides until they form the three-folded structures in the direction of external force. It is because the strong confinement not only strongly reduces the buckling wavelength, but also increases the critical buckling force threshold.

For the weakly confined polymers in the de Gennes regime, the linear polymer shrinks into the condensed states in the compression process under $f/k_{B}T=0.01$. However, the ring polymer shrinks to a length much greater than the linear polymer, due to the excluded volume interactions of two strands in the crowded environment. For $f/k_{B}T=10.0$, the compression force overwhelms the bending energy and excluded volume interactions; thus, both the linear and ring chains collapse into the three-folded structures.

The stretching deformations of ring and linear polymers are the synergistic effects of confinement and stretching forces. These findings are important for understanding the complex compression and stretching of linear and ring polymers with different rigidities in mechanical compression or driven transport. This work is similar to the case that microtubules embedded in a matrix formed by the rest of the cytoskeleton. When microtubules are compressed, the high-amplitude lateral deflections are suppressed by the surrounding matrix, and they buckle [27]. The compression process of a confined polymer also provides detailed information for understanding genome packaging into a procapsid.

## Figures and Tables

**Figure 1 polymers-13-04193-f001:**
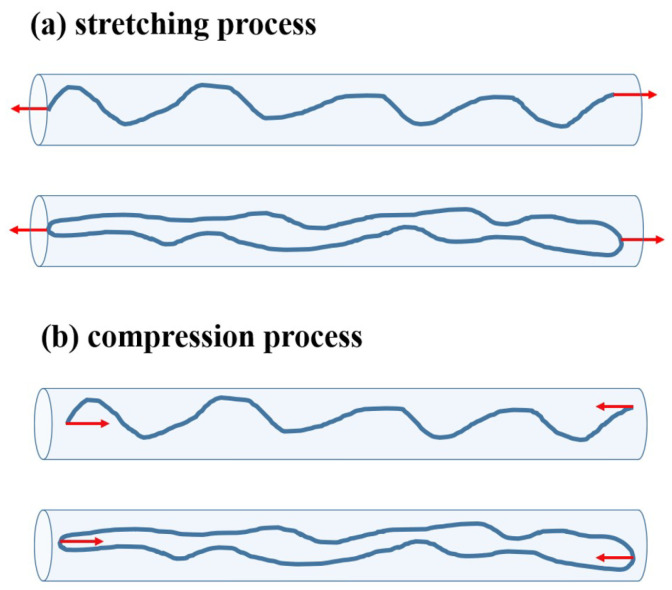
Sketchs of (**a**) the stretching processes and (**b**) compression processes for linear and ring polymers, caused by external forces on two ends or two sides, respectively.

**Figure 2 polymers-13-04193-f002:**
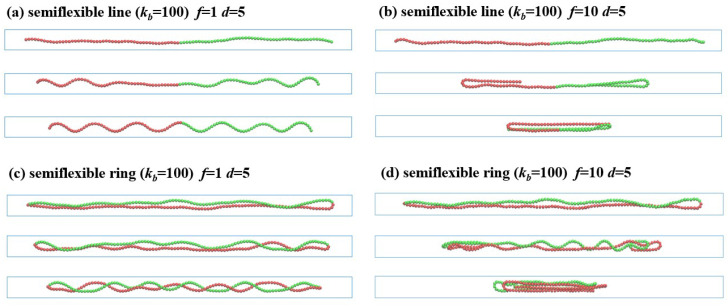
Compression of semiflexible polymers in a confined channel with diameter d=5. (**a**) Linear chain under $f/k_{B}T=1.0$ with the end-to-end distance $L_{x}=97.9,\;89.1,\;81.0$ (from top to bottom). (**b**) Linear chain under $f/k_{B}T=10.0$ with $L_{x}=98.0,\;54.5,\;32.0$. (**c**) Ring chain under $f/k_{B}T=1.0$ with $L_{x}=97.7,\;93.9,\;88.3$. (**d**) Ring chain under $f/k_{B}T=10.0$ with $L_{x}=97.3,\;60.1,\;35.1$. The beads are labeled with red and green for clarity.

**Figure 3 polymers-13-04193-f003:**
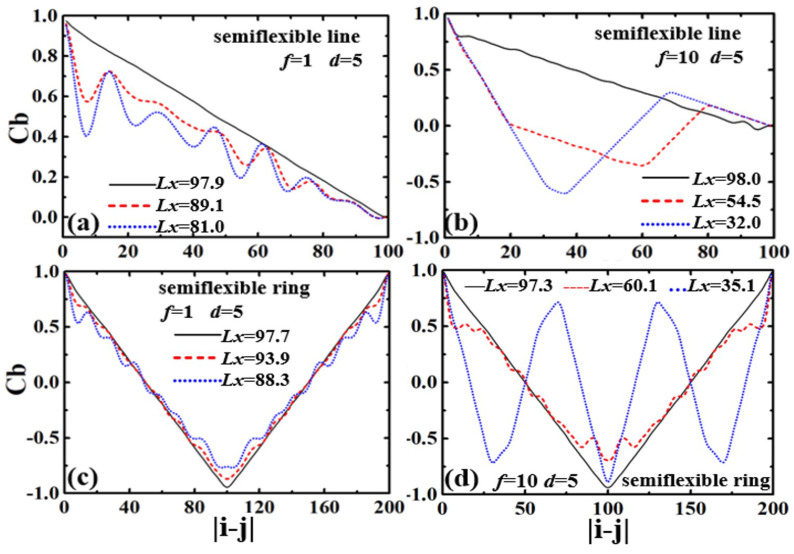
Bond correlation functions $C_{b}$ of confined semiflexible chains in the compression processes: (**a**) linear polymer within f=1 and d=5; (**b**) linear polymer within f=10 and d=5; (**c**) ring polymer within f=1 and d=5; and (**d**) ring polymer within f=10 and d=5.

**Figure 4 polymers-13-04193-f004:**
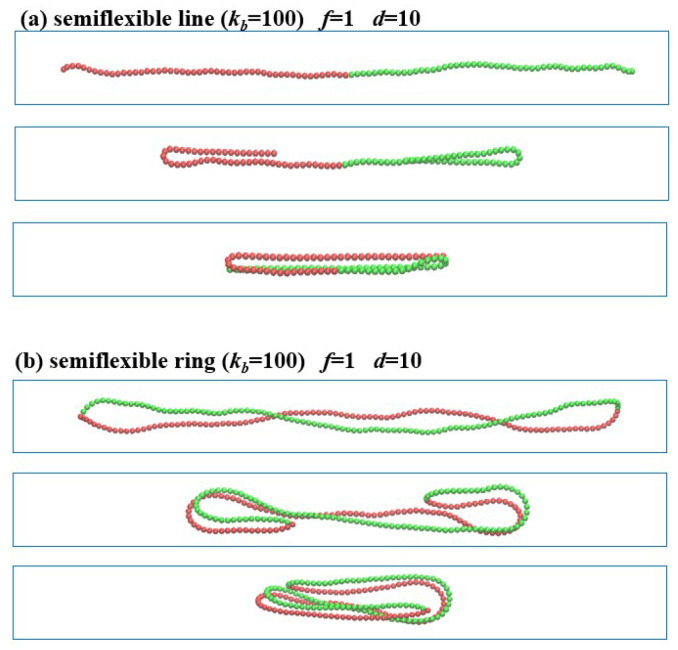
Compression of semiflexible polymers in the confined channel with d=10 under the compression force ($f/k_{B}T=1.0$). (**a**) Linear polymer with the end-to-end distance $L_{x}=97.5,\;57.4,\;31.1$ (from top to bottom). (**b**) Ring polymer with $L_{x}=95.8,\;56.4,\;30.9$.

**Figure 5 polymers-13-04193-f005:**
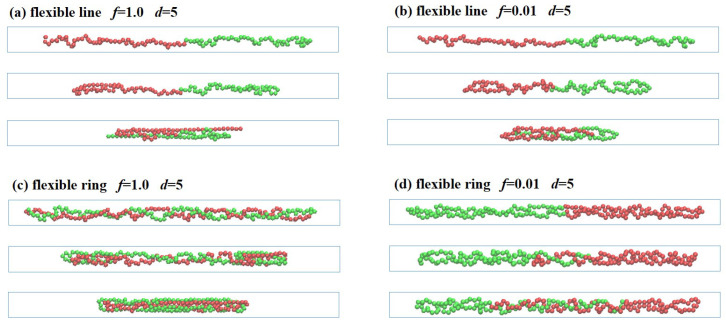
Compression of polymer chains in a confined channel with diameter d=5: (**a**) flexible linear polymer under ($f/k_{B}T=1.0$) with the end-to-end distance $L_{x}=58.1,\;41.7,\;27.8$ (from top to bottom), (**b**) flexible linear polymer under ($f/k_{B}T=0.01$) with $L_{x}=58.6,\;42.5,\;26.5$, (**c**) flexible ring under $(f/k_{B}T=1.0$) with $L_{x}=64.4,\;41.3,\;29.1$, and (**d**) flexible ring under $(f/k_{B}T=0.01$) with $L_{x}=64.2,\;53.8,\;54.6$. The beads are labeled with red and green for clarity.

**Figure 6 polymers-13-04193-f006:**
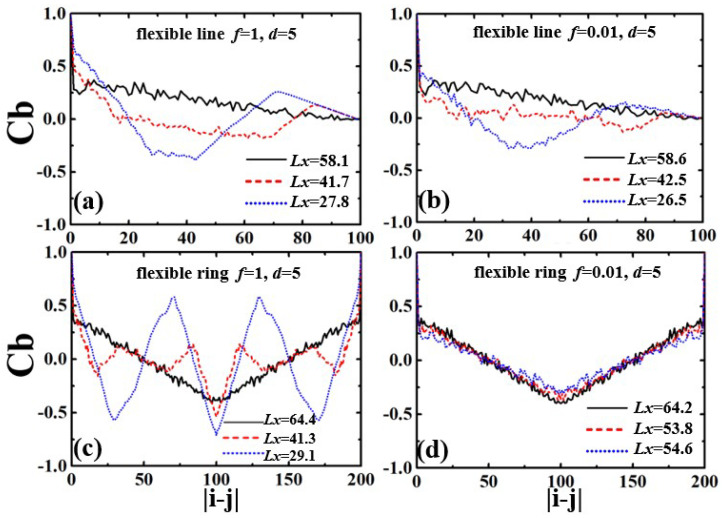
Bond correlation functions $C_{b}$ of the confined flexible chains in the compression processes: (**a**) linear polymer within $f/k_{B}T=1.0$ and d=5, (**b**) linear polymer within $f/k_{B}T=0.01$ and d=5, (**c**) ring polymer within $f/k_{B}T=1.0$ and d=5, (**d**) ring polymer within $f/k_{B}T=0.01$ and d=5.

**Figure 7 polymers-13-04193-f007:**
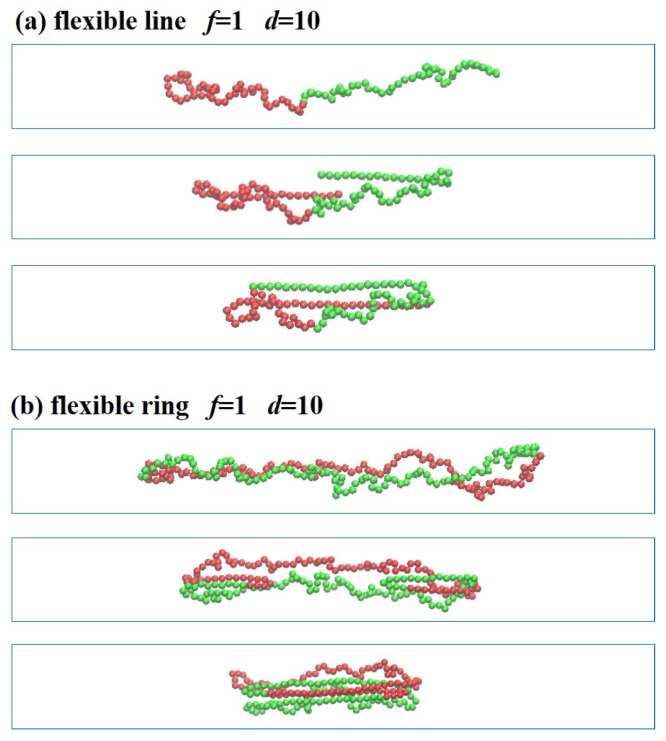
Compression of flexible linear and ring polymers in weak confinement d=10 under (**a**) strong force ($f/k_{B}T=1.0$) with the end-to-end distance $L_{x}=43.1,\;31.7,\;24.5$ and (**b**) weak force $\left(f/k_{B}T=1.0\right)$ with the end-to-end distance $L_{x}=52.3,\;38.9,\;26.6$ (from top to bottom).

**Figure 8 polymers-13-04193-f008:**
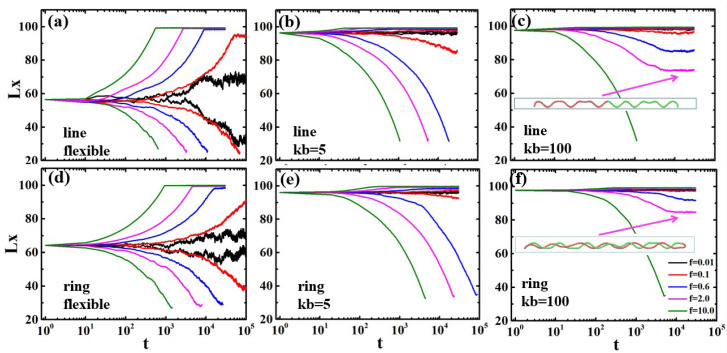
The lengths along the tube Lx as functions of time *t* for linear and ring polymers with different rigidities under external force *f* confined in channel diameter d=5. (**a**) Flexible linear polymer; (**b**) Semiflexible linear polymer with $k_{b}=5$; (**c**) Semiflexible linear polymer with $k_{b}=100$; (**d**) Flexible ring polymer; (**e**) Semiflexible ring polymer with $k_{b}=5$; (**f**) Semiflexible ring polymer with $k_{b}=100$. The insert in (**c**,**f**) is the helix structures for linear and ring polymers, respectively.

**Figure 9 polymers-13-04193-f009:**
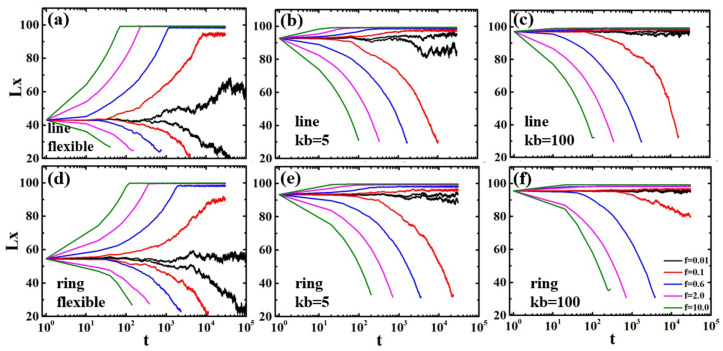
The lengths along the tube $L_{x}$ as functions of time *t* of linear and ring polymers with different rigidities under external force *f* confined in channel diameter d=10. (**a**) Flexible linear polymer; (**b**) Semiflexible linear polymer with $k_{b}=5$; (**c**) Semiflexible linear polymer with $k_{b}=100$; (**d**) Flexible ring polymer; (**e**) Semiflexible ring polymer with $k_{b}=5$; (**f**) Semiflexible ring polymer with $k_{b}=100$.

## Data Availability

We declare that the data supporting the findings of this study are available within the article.

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
