# Peer review of "Compression and Stretching of Confined Linear and Ring Polymers by Applying Force"

_polymers, 2021, doi:10.3390/polym13234193_

Round 1

Reviewer 1 Report

The manuscript "Compression and stretching of confined linear and ring polymers by applying force" by Chen and collaborators is a very detailed and extensive study on the interplay of chain rigidity, degree of confinement and the type of the applied force on the type of compression deformations polymer exhibits.

This study finds its place, especially as fills the gap in the studies on ring polymers, that are more rare and not so systematic in literature (with respect to linear chains), and I do recommend it for publication after some issues are addressed in more detail and the writing style is somewhat improved at few instances.

Although the citing literature seems rather extensive, I did note that the recent article appeared in Polymers in 2017, although very relevant is not being cited in this manuscript. It is the paper "Buckling a Semiflexible Polymer Chain under Compression" by Pilyugina and co-workers. In particular, in this work the authors are reporting that a fluctuating polymer under compressive load requires a larger force to buckle, in the sense that the threshold force required for the onset of the buckling transition is proportional to the degree of fluctuations. Also, they note that "the nature of the buckling transition exhibits a marked change from being distinctly second order in the absence of fluctuations to being a more gradual, compliant transition in the presence of fluctuations".

In relation to this I was wondering how the findings of the current manuscript are comparing with the above mentioned relations, and as the Langevin dynamics simulations have given some indications
on the order of the transition the polymer exhibited. Namely, the onset of helical conformations from purely symmetry point of view would suggest that transition becomes the first order. Please, I would appreciate some clarification on this.

Also, please, a comment/question on the particular choosing on the model for the polymer chain. For the Lennard-Jones potential, the authors say that is inspired by excluded volume, that should be in a broad sense, that of Asakura-Oosawa potential, that should I think depend on the polymer
beads concentration in the solvent, while the LJ potential to me gives more associations with sort of overall attractive (solvo-phobic like interactions), so if you could briefly comment on how author's conclusions would be affected on the polymer model.

As for the style I have noted some points that could be improved (although I am not the native English speaker, and I really think the author's text is of high quality, just some points that I have noted):

1) Line 12 and 13:
"combined actions of polymer rigidity, degree of confinement, and applying force"

could be for example changed to:

"result as an interplay between polymer rigidity, degree of confinement, and applying force"

2) Lines 13,14 and 15:
"When the applying force are beyond the threshold of buckling transition, the semiflexible chain in the strong confinement are firstly buckling and followed by a helical deformation."

could be for example:

"When the applying force is beyond the threshold required for the buckling transition, the semiflexible chain in the strong confinement firstly buckles followed by a helical deformation."

3) Line 109 and 110:

"Finally, we conclude the deformed process and mechanism in the Conclusion section."

could be something like:

"Finally, we draw conclusions on the mechanism and the process of deformations in the Conclusion section."

4) Line 225,226 and 227:
"Compared with the linear polymer, the curve of Cb for the ring polymer has large thermal fluctuations, because of the strong excluded volume between the two strands."

Here I think the terminology is somewhat confusing, since the polymer in question is a ring polymer, so there are not exactly two strands but maybe better to say 'two polymer fragments along the ring...'

Thank you very much in advance for clarifying for me if possible some of the points stated above (since maybe some of the readers might have similar doubts as me)!

Best wishes,
Reviewer

Reviewer 2 Report

This paper presents a study of the deformations of linear and ring polymers in different confinements. A list of issues to be addressed are included here:

  1. The organization and general presentation of the paper should be improved. The current version shows lack of commitment in following instructions and guidelines.
  2. For example, figures should be presented after they are mentioned in the text.
  3. Sometimes the captions of the figures are in the following page.
  4. Section 2 is more a theoretical framework, and the Details on how the simulations were performed are missing.
  5. Units are missing in some expressions or when defining simulations parameters.
  6. which other parameters can be consider in future simulations? can these be related to manufacturing procedures?
  7. Equations 4 and 5 appear in a different format. Additionally, several lines are included in a different font than the one requested by the journal guidelines.
  8. Manuscript should also be proofread for grammar mistakes.
  9. Authors should include in which practical implications are the results obtained useful.

Reviewer 3 Report

The paper presents an extensive analysis of the deformations of linear and ring polymers for different scenarios under stretching and compressive loads applied and both ends.  The polymers are confined in two different diameters and polymer chain lengths. The polymers were modeled by Langevin dynamics.

The paper is well written and organized and the literature review was appropriate. The paper is free from technical errors.

There is no novelty in the paper in terms of analyses. The contribution is solely a detailed study of the polymer mechanics based on known techniques. The results are of interest to researchers in the area.

Since the paper is heavily based on numerical analyses, the authors should:

  1. Include the average computational time for the analyses;
  2. Include an assessment of the accuracy of the results;
  3. Make sure that all data required for the reproduction of the analyses are available.

This reviewer recommends the paper to be accepted after the minor corrections above.
